# Effects of foliar fungicide on yield, micronutrients, and cadmium in grains from historical and modern hard winter wheat genotypes

Hollman Motta-Romero[1], Ferdinand Niyongira[1], Jeffrey D. Boehm, Jr.[2], Devin J. Rose[1,3]*

1 Department of Food Science and Technology, University of Nebraska-Lincoln, Lincoln, NE, United States of America, 2 Wheat, Sorghum and Forage Research Unit, US Department of Agriculture-Agricultural Research Service, Lincoln, NE, United States of America, 3 Department of Agronomy & Horticulture, University of Nebraska-Lincoln, NE, United States of America

* drose3@unl.edu

**Data Availability Statement:** All relevant data are within the paper and its Supporting Information files.

## Abstract

Plant breeding and disease management practices have increased the grain yield of hard winter wheat (*Triticum aestivum* L.) adapted to the Great Plains of the United States during the last century. However, the effect of genetic gains for seed yield and the application of fungicide on the micronutrient and cadmium (Cd) concentration in wheat grains is still unclear. The objectives of this study were to evaluate the effects of fungicide application on the productivity and nutritional quality of wheat cultivars representing 80 years of plant breeding efforts. Field experiments were conducted over two crop years (2017 and 2018) with eighteen hard winter wheat genotypes released between 1933 and 2013 in the presence or absence of fungicide application. For each growing season, the treatments were arranged in a split-plot design with the fungicide levels (treated and untreated) as the whole plot treatments and the genotypes as split-plot treatments in triplicate. The effects on seed yield, grain protein concentration (GPC), micronutrients, phytic acid, and Cd in grains were measured. While the yield of wheat was found to increase at annualized rates of 26.5 and 13.0 kg ha$^{-1}$ yr$^{-1}$ in the presence and absence of fungicide ($P < 0.001$), respectively, GPC (-190 and -180 mg kg$^{-1}$ yr$^{-1}$, $P < 0.001$), Fe (-35.0 and -44.0 µg kg$^{-1}$ yr$^{-1}$, $P < 0.05$), and Zn (-68.0 and -57.0 µg kg$^{-1}$ yr$^{-1}$, $P < 0.01$) significantly decreased during the period studied. In contrast to the other mineral elements, grain Cd significantly increased over time (0.4 µg kg$^{-1}$ yr$^{-1}$, $P < 0.01$) in the absence of fungicide. The results from this study are of great concern, as many mineral elements essential for human nutrition have decreased over time while the toxic heavy metal, Cd, has increased, indicating modern wheats are becoming a better vector of dietary Cd.

## Introduction

Wheat (*Triticum aestivum* L.) is currently grown on a greater land area than any other crop. In 2018 alone, over 214 million hectares were harvested worldwide, with the United States as the

**Funding:** This project was supported by the Nebraska Agricultural Experiment Station with funding from the Hatch Multistate Research capacity funding program (Accession Number 224073) from the USDA National Institute of Food and Agriculture. This project was also funded through USDA in-house funds appropriated to USDA-ARS CRIS project number 3042-21000-033-00D (Improved Winter Wheat Disease Resistance and Quality through Molecular Biology, Genetics, and Breeding). There was no additional external funding received for this study. The funders had no role in study design, data collection and analysis, decision to publish, or preparation of the manuscript.

**Competing interests:** The authors have declared that no competing interests exist.

fourth-largest producer, only behind India, Russia, and China [1]. In the United States, the Great Plains, which include the Dakotas, Nebraska, Kansas, Oklahoma, and parts of Montana, Wyoming, Colorado, New Mexico, and Texas, are a major wheat-producing region, accounting for the 30% and 50% of the annual production of all and winter wheat, respectively [2].

Plant breeding has offered a valuable tool to meet the demand for wheat from the constantly growing global population and face the challenges posed by climate change. Since the Green Revolution starting in the early 1960s, plant breeders have successfully improved the yield of wheat through enhanced agronomic practices (e.g., fungicide application) and by selecting for desirable traits such as short stature [3], adaptation to abiotic stress, and disease resistance [4]. Plant breeders estimate genetic yield gains over time by simultaneously evaluating modern and old cultivars in their adapted environments in replicated and randomized field trials. Upon harvest, linear regression analyses are generally performed to assess significant trends in yield and other variables over time. In this manner, the productivity of hard winter wheat cultivars adapted to the Great Plains of the United States has been estimated to improve at an annualized rate of 10 to 18 kg ha$^{-1}$ yr$^{-1}$ due to plant breeding efforts [5–8].

Nevertheless, these genetic gains of grain yield might negatively affect the nutritional quality of wheat. For instance, the grain protein concentration (GPC) of wheat in the Great Plains has been shown to decrease by 10 mg kg$^{-1}$ for every kg ha$^{-1}$ increase in grain yield [9]. Therefore, high yielding modern cultivars generally contain less protein than historical varieties [7, 10].

Besides serving as a primary source of calories, wheat contributes to the dietary intake of essential micronutrients for populations that rely on it as a staple crop. The inadequate intake of micronutrients such as iron (Fe) and zinc (Zn) leads to health complications such as anemia [11] and stunting [12], respectively. The concentrations of Fe, Zn, copper (Cu), and magnesium (Mg) have decreased in modern British wheat cultivars when compared to older cultivars [13]. In the Great Plains of the U.S., decreasing trends of Fe and Zn in grains of modern wheat varieties have also been reported [9, 14]. Reports on the decrease in micronutrient concentrations in modern compared with historical wheats have driven attention towards older varieties as these are recognized for their higher seed nutrition [15–17]. Nevertheless, studies addressing the impact of the application of fungicide, as a standard agronomic practice, on micronutrients in historical and modern wheats are scarce.

Because of its higher concentration of mineral elements, dietary fiber, and potential health benefits, the consumption of whole wheat flour has gained popularity. Nevertheless, antinutritional compounds in the bran of wheat might attenuate some of these health benefits. Phytic acid, the main storage form of phosphorus (P) in grains, concentrates in the aleurone layer and forms insoluble complexes with mineral elements, collectively referred to as phytate, thus decreasing their bioavailability [18–20]. Indeed, the low phytic acid trait is associated with increased bioavailability of mineral elements in wheat flour [21–23]. In one report, this antinutritional compound steadily increased during the last eighty years of crop development in Pakistan [24]; however, there are no studies on the temporal trends of phytate concentration in cultivars derived from the last century of plant breeding in the Great Plains.

Besides its potentially lower nutritional quality, modern wheat might serve as a vehicle of toxic heavy metals. Cadmium (Cd), a heavy metal recognized as a human carcinogen [25], is introduced into cropland as a result of industrial and agronomic activities [26, 27]. Since wheat plants inadvertently accumulate Cd in grains [28, 29], wheat is recognized as a primary source of dietary Cd for the general population [30]. Multiple governments have established maximum permissible levels of Cd in grains (0.1–0.2 mg kg$^{-1}$) to prevent detrimental health effects such as bone demineralization, tumors, and kidney dysfunction [31]. There is one study on the variability of Cd in a panel of historical and modern hard winter wheat from the Great

Plains germplasm. Although high heritability was found for Cd, the relationships between registration year and the concentration of grain Cd was not evaluated [9].

Extensive research on the dynamics of Cd in durum wheat have led to the reduction of grain Cd concentrations in modern cultivars. The discovery of the gene *Cdu1*, controlling the root-to-shoot Cd translocation in durum wheat, has allowed breeders to introduce the low-Cd trait in modern varieties [32]. Additionally, several studies have shown a negative correlation between plant height and grain Cd. As the leaves and stems compete as Cd sinks during grain filling, less Cd reaches the grains in taller wheat cultivars [28, 33]. Nevertheless, the molecular basis of this relationship is yet to be determined.

The understanding of the dynamics of Cd in bread wheat is in a more premature state than that in durum wheat. A homoeologous locus to *Cdu1* in bread wheat largely explaining the grain Cd variation across cultivars has not been identified [34]. Although negative correlations between plant height and grain Cd have also been reported, only an infrequent reduced height gene (Rht8) has been associated to grain Cd [3, 35, 36]. In a genetic mapping study, a wheat recombinant inbred line (RIL) population (n = 105) segregating for grain Cd concentration levels was used to identify alleles with increased Cd concentrations. Results revealed that RILs possessing the GWM261 allele on chromosome (Chr) 5A (conditioning shorter parenchyma cells in internode and peduncle) and the heavy metal accumulating *HMA3* allele (conditioning elevated Cd concentrations) also on Chr 5A had the highest Cd concentrations among all RILs evaluated in the study, with additive effects for elevated Cd levels [36]. The GWM261 locus is tightly linked (0.6 cM) to the reduced height *Rht8* locus [37]. Therefore, breeders using the *Rht8* locus to select for reduced height and resistance to lodging were advised to concurrently select for low-Cd alleles at the *HMA3*-linked locus on Chr 5A to develop new dwarfing varieties with lower seed Cd concentration [36].

In addition to plant breeding, agronomic practices, such as the application of fungicides, could also affect the concentration of micronutrients and Cd accumulation in wheat grains. On the one hand, fungicide could decrease mineral element concentrations because of increased yields. On the other hand, higher concentrations of elements could occur if the application of fungicide extends the grain filling stage, thereby enabling more mineral elements to be deposited into the grains [38].

The objectives of the present study were to estimate genetic gains and historical trends for important economic (yield and GPC) and nutritional (grain mineral elements, phytate, and Cd) traits of hard winter wheat by evaluating a historic set of eighteen wheat cultivars and two landraces adapted for production in the Great Plains in the presence and absence of fungicide treatment.

## Materials and methods

### Plant materials and experimental design

Twenty hard winter wheat cultivars, including two landraces and eighteen elite cultivars, released or registered between 1870 and 2013 were included in this study (Table 1). These cultivars, originating from the U.S. states of Kansas, Nebraska, Texas, and Oklahoma, were selected based on their historical relevance for grain production, known adaptation of climate conditions of the location of the study, and their contribution to the pedigrees of modern genotypes widely grown in the Great Plains [7, 9]. Nineteen out of the twenty genotypes evaluated herein are present in the hard winter association mapping panel, and their genotyping data is publicly available [9]. The two landraces were included as references of the improvements of elite cultivars due to plant breeding. Because among the elite cultivars only 'Wesley' was found to carry the dwarfing *Rht8* allele related to grain Cd, comparisons among genotypes based on their plant height were not pursued.

**Table 1. Cultivar, registration year, region of origin, Plant Introduction (PI) or Cultivar Introduction (CI) number, and level of resistances against foliar diseases (leaf and stripe rust) for the set of historic wheat genotypes evaluated in the present study[*].**

| ID | Cultivar | Registration year | Region of origin | PI or CI number | Leaf rust[b] | Stripe rust[b] | Source |
|---|---|---|---|---|---|---|---|
| 1 | Turkey[a] | 1874 | Russia | CI 12137 | S | S | [39] |
| 2 | Kharkof[a] | 1900 | Ukraine | PI 5641 | MS | S | [39] |
| 3 | Cheyenne | 1933 | Nebraska | CI 8885 | S | S | [39] |
| 4 | Red Chief | 1940 | Kansas | CI 12109 | S | S | [39] |
| 5 | Wichita | 1944 | Kansas | CI 11952 | S | S | [39] |
| 6 | Warrior | 1960 | Nebraska | CI 13190 | S | S | [40] |
| 7 | Lancer | 1963 | Nebraska | CI 13547 | S | S | [41] |
| 8 | Triumph 64 | 1964 | Oklahoma | CI 13679 | S | S | [42] |
| 9 | Sturdy | 1966 | Texas | CI 13684 | S | S | [43] |
| 10 | Scout 66 | 1967 | Nebraska | CI 13996 | MS | S | [44] |
| 11 | Clark's cream | 1972 | Kansas | PI 476305 | S | S | [45] |
| 12 | Centurk 78 | 1978 | Nebraska | CI 17724 | S | S | [46] |
| 13 | Centura | 1983 | Nebraska | PI 476974 | MS | S | [47] |
| 14 | TAM 107 | 1984 | Texas | PI 495594 | S | S | [48] |
| 15 | Wesley | 1998 | Nebraska | PI 605742 | S | S | [49] |
| 16 | Jagalene | 2002 | Kansas | PI 631376 | S | S | [50] |
| 17 | Anton | 2008 | Nebraska | PI 651044 | MR | S | [51] |
| 18 | Overland | 2007 | Nebraska | PI 647959 | MR | MR | [52] |
| 19 | Settler CL | 2009 | Nebraska | PI 653833 | MS | MS | [53] |
| 20 | Freeman | 2013 | Nebraska | PI 667038 | S | MR | [54] |

[a] Landraces.

[b] The level of resistance towards leaf and stripe rust is reported in the scale, S: susceptible, MS: moderately susceptible, MR: moderately resistant, R: resistant.

[*] The registration years were obtained from the Genetic Resources Information and analytical System (GRIS) for wheat and triticale.

For each of two growing seasons, 2017 (2016–2017) and 2018 (2017–2018), the experimental design consisted of a split-plot design with main plots in randomized complete blocks of three replications. The whole plot treatments were two levels of fungicide (treated and untreated). A single application of a broad-spectrum fungicide (Caramba, BASF, NJ, USA) was performed at the booting stage at the dose rate recommended by the manufacturer to simulate the common agronomic practice of the region [55]. The twenty cultivars were planted in 1 m$^2$ plots within each block as the split-plot treatments. In 2017, each plot was visually evaluated for foliar disease severity two weeks after fungicide intervention. To report plant health, the flag leaf was scored from 0 to 10, where 0 represented a highly damaged leaf and 10 a completely healthy green leaf. 2018 data were not recorded, thus are not reported in this manuscript. The grain yield (kg ha$^{-1}$) for each cultivar was determined by harvesting the full plot with a simple plot combine harvester. The diameter and thousand kernels dry weight (TKW) of the grains were determined using a single-kernel characterization system 4100 (Petern Instruments, Huddinge, Sweden).

This research was carried out at the University of Nebraska Eastern Nebraska Research and Extension Center (ENREC) near Mead, NE USA. In an effort to create measurable variability in the grain Cd concentration among the samples, this location was selected because of its known elevated concentrations of soil Cd [8, 34]. Depending on soil properties such as organic matter content and pH, the soil diethylene triamine pentaacetic acid (DTPA) extractable Cd at this location was reported to range between 0.16 to 0.26 mg kg$^{-1}$, whereas 0.04 mg kg$^{-1}$ have been reported for surrounding locations [8]. The high heavy metal concentration of this land

is thought to be due to its use by the U.S. military for munitions manufacturing during the World War II era.

## Mineral element analysis

Mineral elements were determined from whole wheat kernels using a wet ashing procedure followed by quantification through inductively coupled plasma mass spectrometry [9]. In brief, 2 g of wheat kernels were washed with 0.3 N hydrochloric acid in a Buchner funnel and then rinsed with distilled water to remove debris and adhering elements from the surface of the kernels. The samples were then predigested overnight in 3.5 mL of concentrated nitric acid at room temperature and then digested by incubating for 1 h at 40˚C followed by 1 h at 100˚C in a BD50 digestion block (Seal-Analytical, WI, USA). Subsequently, 3.5 mL of nitric acid were added, and the tubes were incubated for another 1 h at 100˚C. Next, 4 mL of 30% hydrogen peroxide were added and the samples allowed to incubate at 125˚C for 1.5 h followed by another dose of 4 mL of hydrogen peroxide and an additional 1 h at 125˚C. The samples were then evaporated to dryness by raising the temperature to 150˚C for 2 h and then 165˚C until the sample appeared dry.

The dried ash was resuspended in 12 mL of 1% nitric acid containing 50 ppb gallium as an internal standard and mineral elements were analyzed using a inductively coupled plasma mass spectrometer (7500cx ICP-MS Agilent Technologies Inc., CA, USA) operating in kinetic discrimination mode with helium gas at 5 mL min$^{-1}$. The injection volume was 40 μL; each sample replicate was measured in duplicate, and 2% nitric acid was used to rinse between samples. The results obtained for Cd, Fe, Mg, Mn, P, and Zn are presented in this research.

## Wheat milling, protein, and phytate concentration of whole wheat flour

Prior to milling, wheat kernels were dried to approximately 8% moisture content in a convection oven at 40˚C overnight [56]. The milling was performed using a laboratory roller mill (Quadrumat Jr, CW Brabender Instruments Inc., South Hackensack, NJ, USA) with the sifting screen removed to recover the resulting whole wheat flour. For the determination of protein and phytate concentrations, the flour was manually mixed before sampling to evenly distribute the bran particles.

The GPC of whole wheat flours was measured using a nitrogen analyzer (FP 528, LECO, St. Joseph, MI, USA) with a protein conversion factor of 5.7.

Phytate was measured as previously reported [57, 58]. In brief, phytic acid was extracted from 250 mg of whole wheat flour in 10 mL of 0.2 N hydrochloric acid by shaking overnight in a 4˚C water bath. The contents were centrifuged (3000 g, 5 min), the supernatant diluted to a final volume of 25 mL with distilled water, and the pellet discarded. Next, 0.25 mL of the extract was mixed with 0.75 mL of 0.2 N hydrochloric acid, and 1.0 mL of 450 μM ammonium iron (III) sulfate and placed in a boiling water bath for 30 min. Subsequently, the tubes were cooled and centrifuged (3000 g, 5 min). One mL of the supernatant was mixed with 1.5 mL of 2,2′-bipyridine, and the absorbance was measured at 530 nm. A standard curve of sodium phytate dodecahydrate (containing 16% phytate phosphorus measured using ICP-MS as described for the mineral element analysis above) was used to quantify the concentration of phytate in the samples.

## Statistical analyses

The data from both growing years were analyzed using a three-factor factorial ANOVA. Registration year (as a continuous variable), fungicide, growing year, and their interactions were

treated as fixed effects, whereas replications nested within growing year was treated as a random effect. Because of the significant effect of growing year on multiple response variables, data were analyzed within each individual growing year to evaluate the effects of registration year and fungicide in each growing year. A correlation analysis between the data from each growing year was performed to evaluate whether the independent variables followed similar temporal trends in both growing years despite differences in their magnitude.

To visualize the relationships among the independent variables and registration year, all variables were adjusted for growing year to account for its magnitude effect. In brief, each independent variable was regressed on growing year (as continuous variable) and the residuals saved. To obtain values in a practical range, the mean of the raw data was added to the residuals. Finally, these adjusted variables were regressed on registration year.

To evaluate the dilution effect of the increasing yield on the historical trends of the response variables, a partial correlation analysis, accounting for the effect of growing year and yield, was performed for variables that showed significant regressions with registration year. As mentioned above, in this work the landraces 'Turkey' (1874) and 'Kharkof' (1900) were included as references of the improvement of elite cultivars due to plant breeding. However, these landraces were not included in the regression analyses due to the outlying nature of their release years in the context of the period studied (1933–2013).

A principal component analysis was performed to determine the influence of yield on the effect of fungicide on grain micronutrient concentrations. This data analysis was performed using R-studio (version 1.2.5019–6).

## Results

### Environmental conditions differed between the two growing years, but economic and nutritional traits of wheat presented similar temporal dynamics in both growing years

First, the weather conditions from the 2017 and 2018 growing seasons differed considerably (Fig 1). 2017 was considered a good year in terms of yield in the region. Dry conditions allowed for planting during early October. Due to the wet conditions in April and May, stripe rust (*Puccinia stritiformis* Westend) and powdery mildew (*Blumeria graminis* f. sp. *tritici*) were the major diseases observed in 2017 [59], whereas the warm dry conditions that followed during flowering from late May to mid-June prevented fusarium head blight (*Fusarium graminearum*) during this growing season [60]. On the other hand, 2018 was an abnormal year. High precipitation in early October delayed planting until the last week of the month. The unusually dry and cold weather in November, combined with late planting, reduced tillering and negatively affected yield. The region also experienced the coldest April and warmest May on record. Due to low precipitation, there was little disease pressure during the spring [61]. However, a rainy June allowed for higher disease pressure after flowering with leaf rust (*Puccina triticina*), bacterial leaf streak (*Xanthomonas translucens*), and fusarium head blight (*Fusarium graminearum*) as the main diseases observed [62].

Growing year, as a main effect or interacting with fungicide or registration year, had a significant effect on all the economical and nutritional traits studied (Table 2). This is likely due to the considerable differences in environmental conditions between the two growing years, as discussed. Nevertheless, the positive correlations between all the response variables across the two growing years (S1 Table), except for Mg in the presence of fungicide, indicated that despite the significant effect of growing year, similar trends were observed in both growing seasons under each fungicide treatment. This suggests changes in magnitude of the effects due to environment rather than crossover effects. To account for this magnitude effect in the analysis of

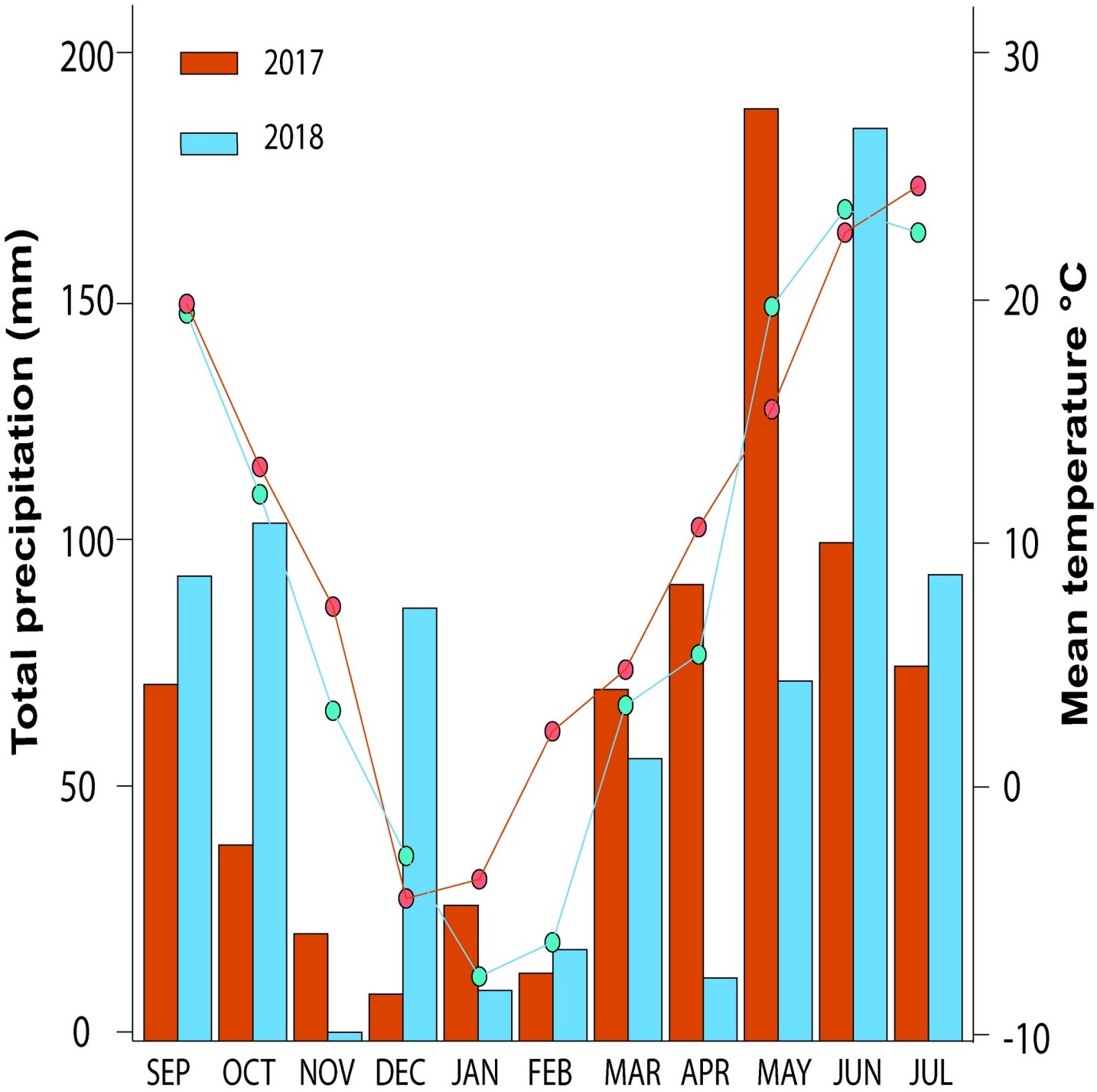

**Fig 1. Monthly total precipitation (bars) and average temperature (lines) during the 2017 (2016–2017) and 2018 (2017–2018) winter wheat growing seasons at the Eastern Nebraska Research and Extension Center (ENREC) near Mead, NE, USA, for a historic set of 20 wheat cultivars or landraces released between 1873 and 2013, which were grown in a split-plot field experiment in the presence or absence of fungicide application.**

fungicide treatment, the response variables were partially regressed on registration years, while controlling for the effect of growing year under each fungicide treatment for further analysis.

**Table 2. Analysis of variance (F values) for grain yield and grain protein concentration (GPC), phytate, magnesium (Mg), phosphorus (P), manganese (Mn), iron (Fe), zinc (Zn), cadmium (Cd) concentrations, thousand kernel weight (TKW), and grain diameter for a historic set of 18 wheat cultivars released between 1933 and 2013, which were grown in a split-plot field experiment in the presence or absence of fungicide application at the Eastern Nebraska Research and Education Center in 2017 and 2018.** Results are presented for the full model (both growing years) and for each growing year (2017 and 2018).

| ANOVA F values for full model | | | | | | | | | | | |
|---|---|---|---|---|---|---|---|---|---|---|---|
| Effect | df | Yield | GPC | Phytate | Mg | P | Mn | Fe | Zn | Cd | TKW | Diameter |
| Registration Y (Reg) | 1 | 98.7*** | 85.5*** | 7.0** | 0.2 | 6.8** | 0.7 | 12.3*** | 24.5*** | 31.6*** | 0.55 | 0.2 |
| Fungicide (Fun) | 1 | 64.2*** | 40.6*** | 248.7*** | 226.7*** | 79.9*** | 250.6*** | 123.1*** | 78.8*** | 81.3*** | 40.1*** | 68.4*** |
| Growing year (GY) | 1 | 2.5 | 652.2*** | 54.0*** | 2.8 | 5.1* | 9.2** | 384.8*** | 84.0*** | 52.6*** | 25.4*** | 1.19 |
| Reg * Fun | 1 | 9.1** | 0.3 | 0.0 | 0.0 | 0.3 | 0.0 | 0.1 | 0.0 | 4.5 | 3.3 | 3.36 |
| Reg * GY | 1 | 39.4*** | 5.9* | 0.2 | 1.2*** | 0.0 | 0.9 | 0.0 | 0.9*** | 0.5 | 0.1 | 2.9 |
| Fun * GY | 1 | 28.2*** | 19.8*** | 2.2 | 11.7 | 1.9 | 2.5 | 0.0 | 14.0 | 2.7 | 16.7*** | 31.8*** |
| Reg * Fun * GY | 1 | 7.9** | 4.2* | 1.8 | 0.5 | 0.1 | 0.1 | 0.2 | 0.1 | 0.6 | 0.32 | 0.6 |
| Residuals | 232 | | | | | | | | | | | |
| ANOVA F values for 2017 | | | | | | | | | | | |
| | df | Yield | GPC | Phytate | Mg | P | Mn | Fe | Zn | Cd | TKW | Diameter |
| Reg | 1 | 82.1*** | 82.3*** | 4.7* | 0.9 | 3.6 | 0.0 | 6.3* | 16.5** | 14.8*** | 0.15 | 0.6 |
| Fun | 1 | 55.4*** | 2.0 | 143.0*** | 131.3*** | 26.8*** | 145.2*** | 70.0*** | 76.9*** | 34.0*** | 46.4*** | 80.9*** |
| Reg * Fun | 1 | 10.6** | 1.4 | 0.7 | 0.3 | 0.3 | 0.0 | 0.4 | 0.0 | 5.1 | 2.5 | 2.8* |
| Residuals | 116 | | | | | | | | | | | |
| ANOVA F values for 2018 | | | | | | | | | | | |
| | df | Yield | GPC | Phytate | Mg | P | Mn | Fe | Zn | Cd | TKW | Diameter |
| Reg | 1 | 16.8*** | 19.4*** | 2.5 | 0.3 | 3.3 | 1.7 | 6.0* | 8.5** | 6.0* | 0.6 | 2.9 |
| Fun | 1 | 9.1** | 48.6*** | 108.1*** | 97.7*** | 58.0*** | 106.7*** | 58.1*** | 14.0*** | 58.1*** | 3.1 | 4.5* |
| Reg * Fun | 1 | 0.1 | 2.8 | 1.2 | 0.2 | 0.0 | 0.1 | 0.0 | 0.2 | 0.0 | 1.0 | 0.4 |
| Residuals | 116 | | | | | | | | | | | |

*Significant at the 0.05 probability level.

**Significant at the 0.01 probability level.

***Significant at the 0.001 probability level.

## Differential temporal dynamics and benefits of fungicide on the economic traits of wheat

The economic traits of seed yield and GPC were significantly affected by registration year and fungicide application ($P < 0.001$, Table 2). The yield of wheat increased, while its GPC decreased over the registration years studied. The yield of wheat increased at a rate of 26.5 kg ha$^{-1}$ yr$^{-1}$ under fungicide application and at 13.0 kg ha$^{-1}$ yr$^{-1}$ in the absence of fungicide since 1933 (Fig 2A).

The improvement in yield over time is known as the genetic gain obtained through plant breeding and may also be expressed as a percentage of the yield of a check cultivar or historic landrace grown in the region [63]. In the case of the Great Plains, 'Kharkof' is commonly used as the check cultivar [63–65]. Therefore, the rates obtained in this study are equivalent to 0.86% and 0.47% yr$^{-1}$ over Kharkof in the presence and absence of fungicide since 1933, respectively.

The fungicide treatment presented an interaction with registration year (Fig 2A), yet, in general, the application of fungicide increased the mean yield of all cultivars. The range of yield increased from 3101–4418 kg ha$^{-1}$ to 3685–5794 kg ha$^{-1}$ under the fungicide treatment since 1933. While the landraces Turkey and Kharkof increased from 3094 to 3564 kg ha$^{-1}$ and from 2738 to 3089 kg ha$^{-1}$, respectively.

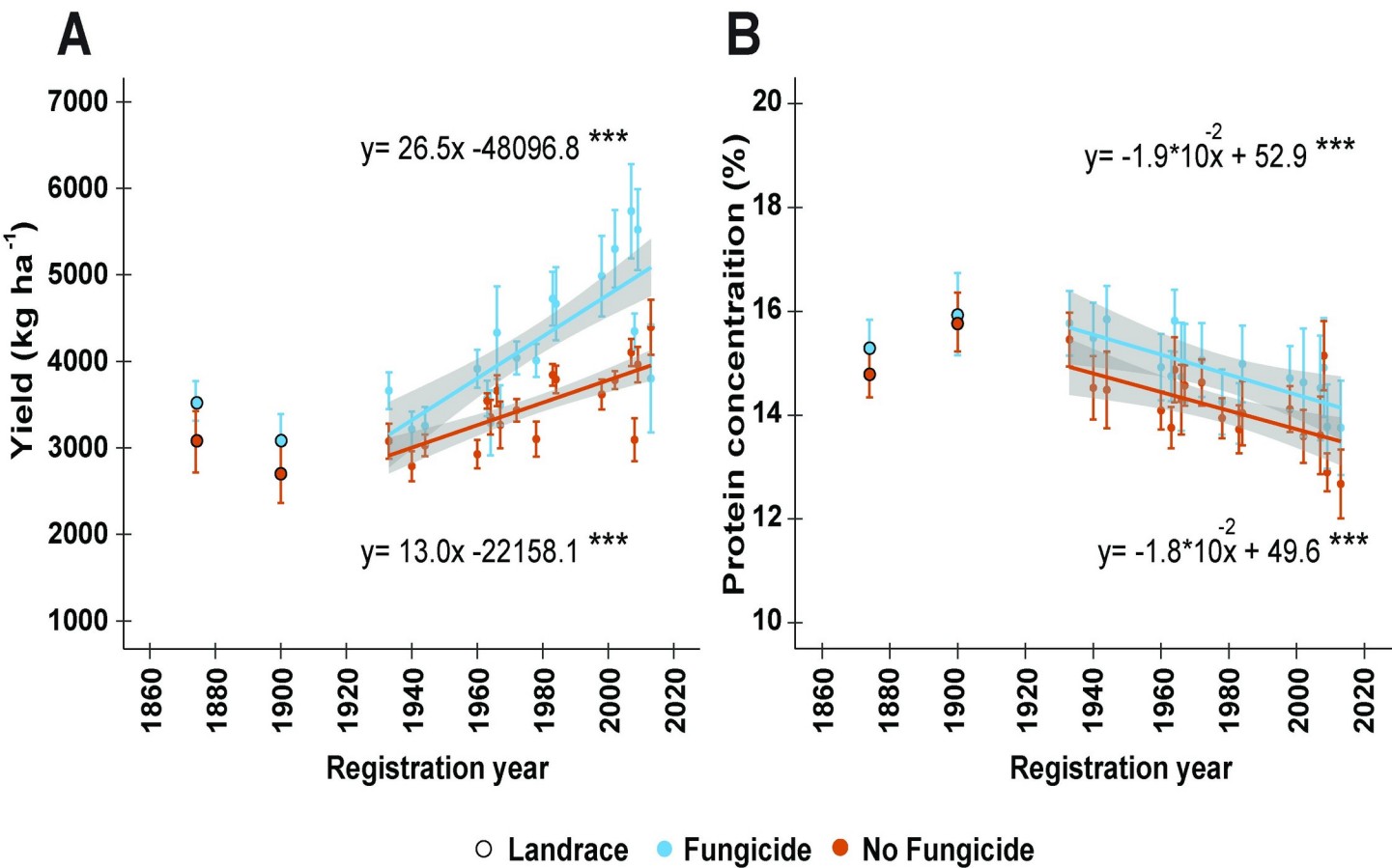

**Fig 2.** Linear regressions of (A) grain yield and (B) grain protein concentration (GPC) by registration year for a historic set of 18 wheat cultivars or landraces released between 1933 and 2013, which were grown in a split-plot field experiment in the presence or absence of fungicide application at the Eastern Nebraska Research and Education Center in 2017 and 2018. The regression lines were adjusted for growing year effects and the shaded area around each regression line represents the 95% confidence interval. Asterisks indicate statistical significance detected at 0.001 probability level. The landraces 'Turkey' (1874) and 'Kharkof' (1900) were not included in the regression analysis.

In 2017, which was characterized by a higher foliar disease pressure during spring, as discussed above, the disease severity was decreased by fungicide in all cultivars, as evidenced by increases in leaf health, except for Kharkof (1900) and 'Lancer' (1963) (S1 Fig). The amount of leaf health improvement differed for all cultivars. Those genotypes with high disease severity such as 'Turkey' (1874), 'Wichita' (1944), 'Warrior' (1960), 'Triumph 64' (1964), 'Centurk 78' (1978), 'Centura' (1983), 'TAM 107' (1984), and 'Settler CL' (2009) had greater benefits from the treatment than less susceptible cultivars.

All the cultivars presented similar grain sizes, including the landraces (S2A and S2B Fig). Registration year presented non-significant regressions with grain diameter or TKW (Table 2). Whereas, fungicide application led to significantly bigger and heavier kernels with means increasing from 2.6 to 2.7 mm of diameter and the TKW from 26 to 28 g (Table 2).

A three-way interaction between registration year, fungicide, and growing year was significant for GPC (Table 2). In general, GPC decreased over the period studied (Fig 2B). The concentration of protein decreased at an annualized rate of 190 and 180 mg kg$^{-1}$ yr$^{-1}$ under the application and absence of fungicide, respectively. The partial correlations between GPC and registration year, when controlling for growing year and yield, remained significant in the

presence ($r$ = -032., $P$ < 0.001) and absence of fungicide ($r$ = -0.30, $P$ < 0.01). The effect of fungicide on GPC was inconsistent, as it presented a significant interaction with growing year, which was only observed in 2018 (Table 2).

## Grain cadmium increased over time in the absence of fungicide while its application attenuated this trend

Grain Cd increased with registration year in the absence of fungicide (Fig 3, Table 2). Most samples, regardless of fungicide treatment, were within 0.1 and 0.2 mg kg$^{-1}$, the maximum permissible limits in China and Europe, respectively. The concentration of grain Cd significantly increased at an annualized rates of 0.4 µg kg$^{-1}$ yr$^{-1}$ in the absence of fungicide ($P$ <0.01). The application of fungicide decreased the range of grain Cd concentrations from 0.1–0.19 mg kg$^{-1}$ to 0.07–0.14 mg kg$^{-1,}$ while no significant temporal trend was observed under fungicide application (Fig 3).

## Phytate and total phosphorus remained constant over time while fungicide had contrasting effects on these components

When controlling for the effect of growing year, the grain phytate concentration remained constant across the registration years studied (Fig 4A).

The application of fungicide led to a significant increase in phytate concentration of all cultivars (P < 0.001; Fig 4A). The overall mean phytate concentration increased from 1.0 mg kg$^{-1}$ to 1.6 mg kg$^{-1}$ in the fungicide treatment.

As phytate is the main storage form of P in wheat grains, grain total P behaved similarly to phytate. Regardless of the fungicide treatment, total P remained constant over time (Fig 4B). However, contrary to phytate, the application of fungicide decreased the concentration of grain P in all cultivars (Fig 4B).

## The temporal dynamics of micronutrients differed while fungicide decreased the concentration of all elements

No significant changes over time were observed for grain Mg and Mn, whereas grain Fe and Zn significantly decreased over the registration years studied (Table 2; Fig 5). Fe decreased at annualized rates of 35 µg kg$^{-1}$ yr$^{-1}$ and 44 µg kg$^{-1}$ yr$^{-1}$ and Zn at 68 µg kg$^{-1}$ yr$^{-1}$ and 57 µg kg$^{-1}$ yr$^{-1}$ in the presence and absence of fungicide since 1933, respectively. When controlling for growing year and yield, while the partial correlations of Fe with registration year became non-significant regardless of the fungicide treatment ($r$ = -0.14 and $r$ = -0.11, $P$ > 0.05), the decreasing trend of Zn remained significant under fungicide treatment (r = -0.29, P < 0.01) and was only attenuated in its absence ($r$ = -0.16, $P$ > 0.05).

To investigate the relationship between genotype susceptibility and mineral elements in grains, the cultivars were grouped into two groups: susceptible, and moderately susceptible and moderately resistant (Table 2). Then, the difference in grain mineral elements between fungicide treatments was compared between these two groups, and no significant differences were found for Cd or any micronutrient ($P$ >0.05).

The application of fungicide resulted in significantly lower concentrations recorded for all micronutrients (i.e., Mg, Mn, Fe, and Zn) and the macronutrient P. In the case of Mg and Mn, which remained constant in the period studied, overall means decreased from 1800 mg kg$^{-1}$ and 41.6 mg kg$^{-1}$ to 1362 mg kg$^{-1}$ and 30.9 mg kg$^{-1}$, respectively. The range of Fe decreased from 35.9–44.5 mg kg$^{-1}$ to 27.50–37.90 mg kg$^{-1}$, whereas that of Zn from 25.5–35.2 mg kg$^{-1}$ to 22.2–30.9 mg kg$^{-1}$.

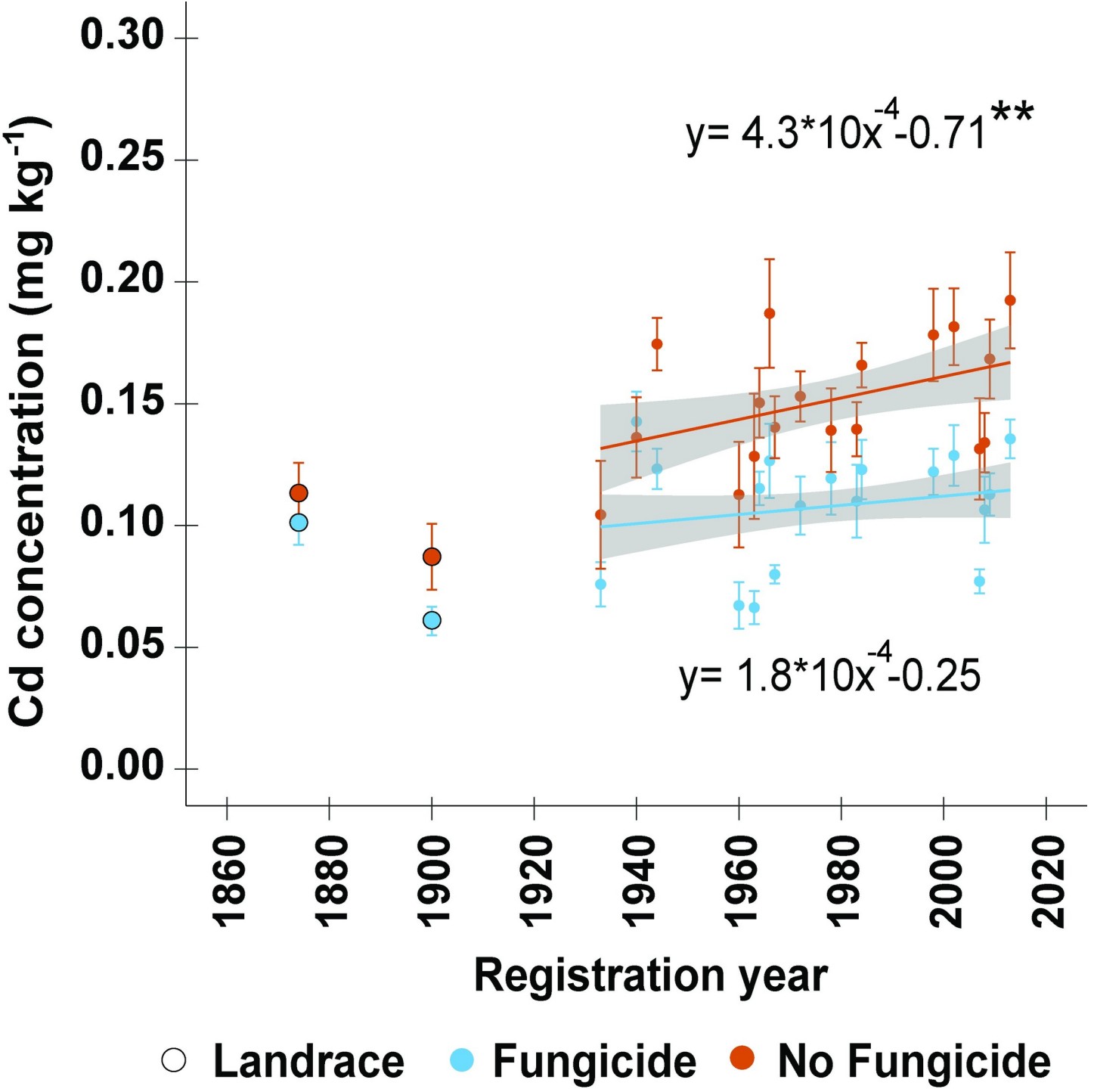

**Fig 3. Linear regression of grain cadmium (Cd) concentration by registration year for a historic set of 18 wheat cultivars or landraces released between 1933 and 2013, which were grown in a split-plot field experiment in the presence or absence of fungicide application at the Eastern Nebraska Research and Education Center in 2017 and 2018.** The regression lines were adjusted for growing year effects and the shaded area around each regression line represents the 95% confidence interval. Asterisks indicate statistical significance detected at the 0.01** and 0.001*** probability levels, respectively. The landraces 'Turkey' (1874) and 'Kharkof' (1900) were not included in the regression analysis.

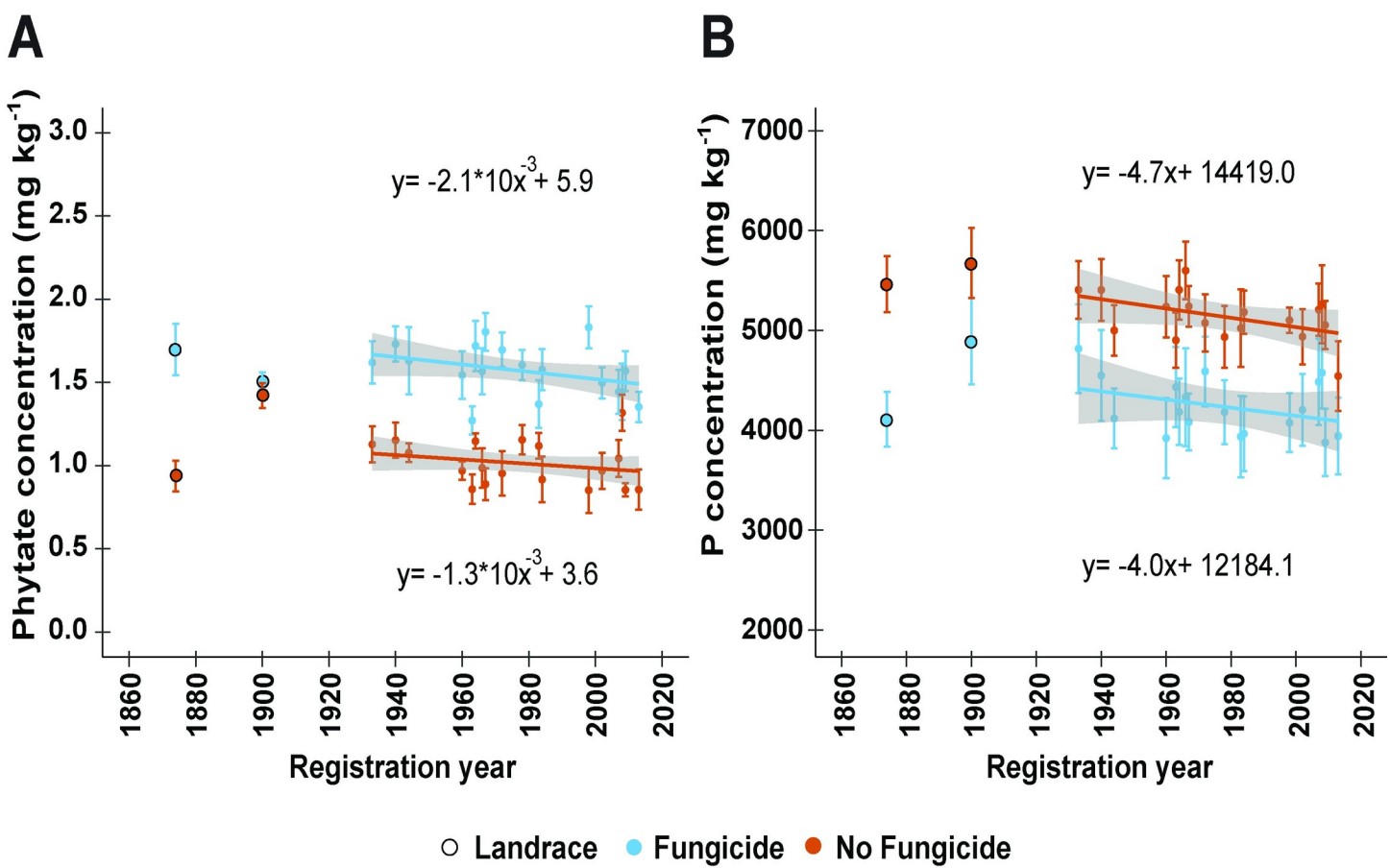

**Fig 4.** Linear regressions of (A) phytate and (B) phosphorus (P) concentrations by registration year for a historic set of 18 wheat cultivars or landraces released between 1933 and 2013, which were grown in a split-plot field experiment in the presence or absence of fungicide application at the Eastern Nebraska Research and Education Center in 2017 and 2018. The regression lines were adjusted for growing year effects and the shaded area around each regression line represents the 95% confidence interval. Asterisks indicate statistical significance detected at the .05*, .01** and 0.001*** probability levels, respectively; NS = non-significant. The landraces 'Turkey' (1874) and 'Kharkof' (1900) were not included in the regression analysis.

## The effect of fungicide on grain mineral elements is largely, but not fully, explained by yield dilution

Due to the contrasting effect of fungicide on the yield and the concentration of mineral elements in wheat grains, it was hypothesized that the effect of fungicide on mineral elements could be attributed to a dilution effect of the higher yield of fungicide treated plants, when compared to their non-treated counterparts. Therefore, two principal components (PC) analyses were performed to identify the variables driving the differences among the fungicide treatments and the effect of yield differences. In the first PC analysis, the seed concentration of each nutrient was used as the input data. In the second PC analysis, the concentration of each nutrient was multiplied by the yield to account for the differences in yield between the fungicide treatments.

In the first analysis using the concentration data the first two PCs explained 76% of the total variance (Fig 6A). PC1 explained 53% of the variation and was positively loaded with phytate, grain diameter, and TKW and negatively loaded with all the micronutrients. For PC2, 23% of the variation was positively loaded with protein and negatively loaded with release year. Seed yield and Cd were equally distributed between the two PCs but were oppositely positioned in

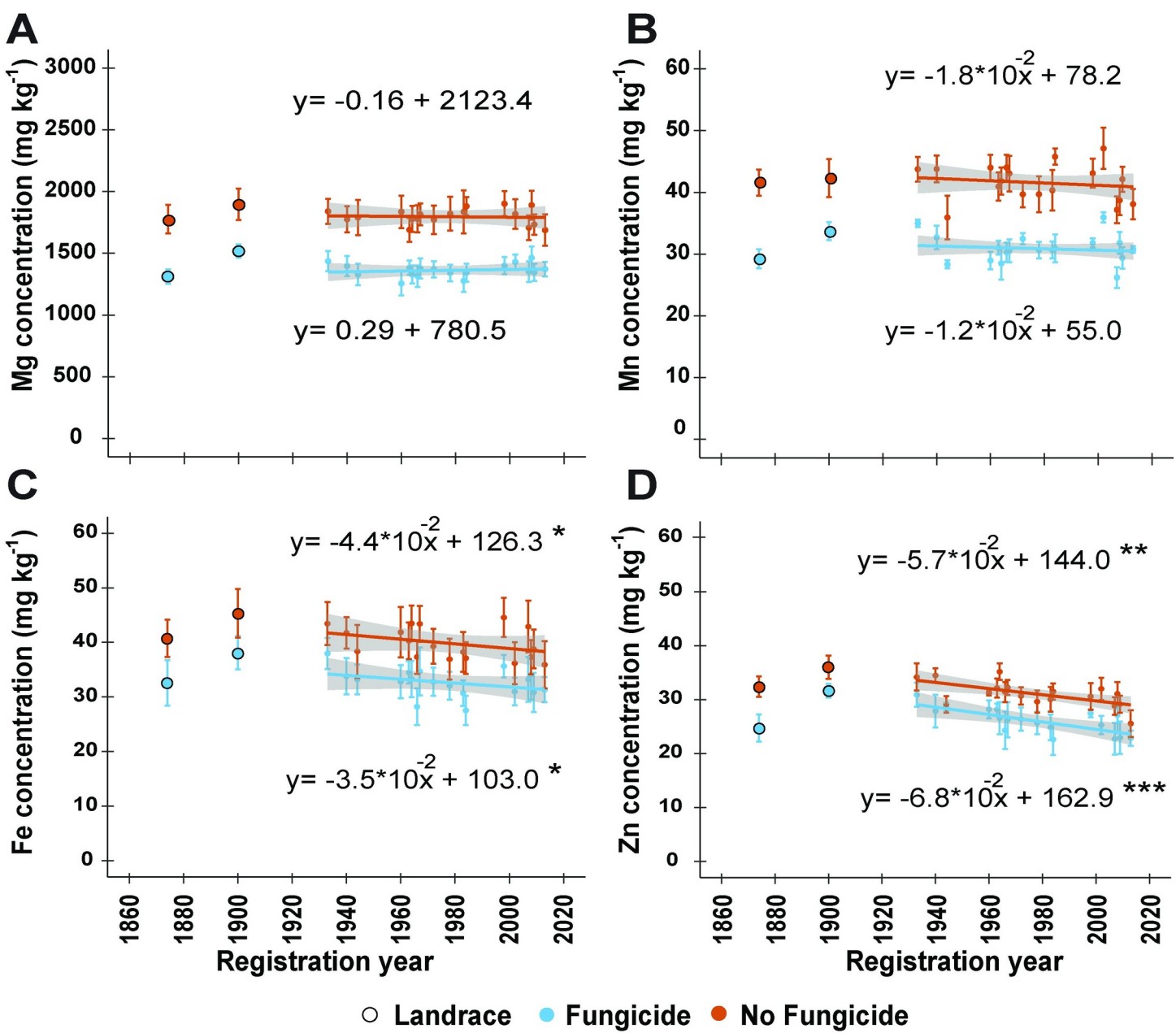

**Fig 5.** Linear regressions of (A) magnesium (Mg), (B) manganese (Mn), (C) iron (Fe) and (D) zinc (Zn) concentrations by registration year for a historic set of 18 wheat cultivars or landraces released between 1933 and 2013, which were grown in a split-plot field experiment in the presence or absence of fungicide application at the Eastern Nebraska Research and Education Center in 2017 and 2018. The regression lines were adjusted for growing year effects and the shaded area around each regression line represents the 95% confidence interval. Asterisks indicate statistical significance detected at the .05*, .01** and 0.001*** probability levels, respectively; NS = non-significant. The landraces 'Turkey' (1874) and 'Kharkof' (1900) were not included in the regression analysis.

PC1. As shown, fungicide treatments were primarily separated along the PC1 axis, where the treated samples had positive Eigenvalues on PC1 and were associated with larger seed sizes, higher yield, GPC, and phytate concentrations, wheres the non-treated group had negative Eigenvalues on PC1 and were associated with higher mineral element concentrations.

In the second principal components analysis where data were expressed relative to yield (Fig 6B), PC1 was mainly loaded with the mineral elements and accounted for 58% of data

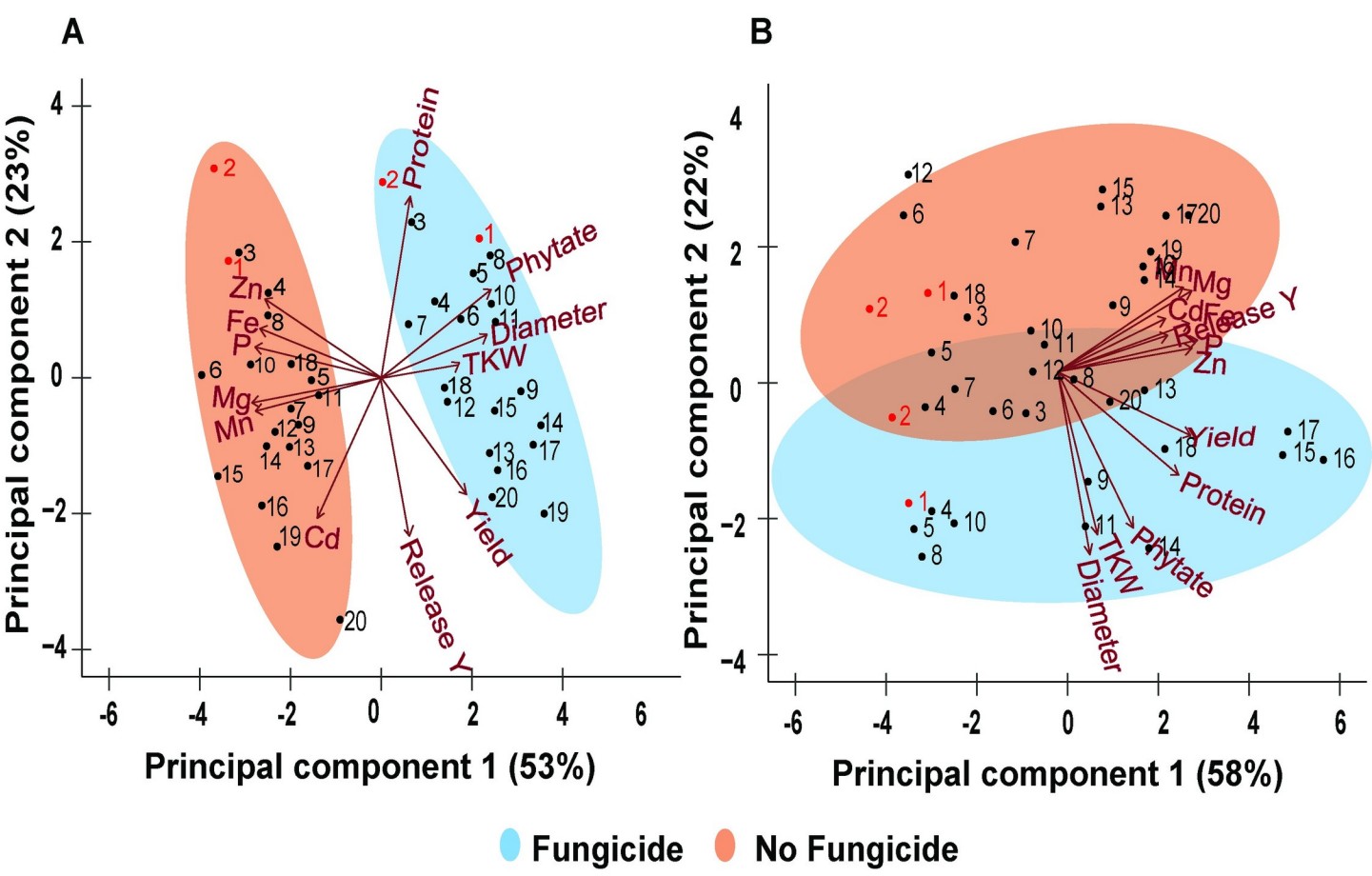

**Fig 6. Principal components analysis, based on correlation matrix, of economic traits (yield and protein), grain diameter, TKW, phytate, micronutrients (Fe, Mg, Mn, P, and Zn), and cadmium (Cd) for a historic set of 18 wheat cultivars or landraces released between 1873 and 2013, which were grown in a split-plot field experiment in the presence or absence of fungicide application at the Eastern Nebraska Research and Education Center in 2017 and 2018.** (A) Vectors are expressed independent of yield. (B) Vectors represent each variable as relative to yield (e.g., mg ha$^{-1}$). The sample ID numbers in (A) and (B) correspond to each wheat cultivar or landrace displayed in Table 1. The landraces 'Turkey' (1) and 'Kharkof' (2) are highlighted in red.

variance, but was no longer involved in separating fungicide treatments. Fungicide treatments were still separated but this time the separation was along the PC2 axis. PC2 only accounted for 22% of the data variance and was heavily loaded with grain diameter, TKW, phytate, and protein.

## Discussion

The yield of hard winter wheat has been significantly improved through plant breeding in the Great Plains. In this study, the genetic gain of yield under fungicide application was estimated at 0.86% yr$^{-1}$ relative to the check cultivar Kharkof. This estimate is within the range (0.4–1.1% yr$^{-1}$) reported for cultivars adapted to the region [5–7, 63, 64]. In the absence of fungicide, the lower genetic gain (0.47% yr$^{-1}$) points to the susceptibility of these cultivars to fungal diseases, whereas the positive relationship between registration year and yield indicates the increasing yield potential of modern wheat under disease pressure. These results contrast with previous studies where higher estimates are reported in the absence of fungicide than upon treatment, mainly due to the steady increase in disease resistance of modern cultivars, whose yield benefits less from the application of fungicides [66, 67]. Nevertheless, in the case of the great plains,

particularly in the period studied (1933–2013), winter wheat has been bred for a wide range of traits besides disease resistance, e.g., yield potential, end-use quality, and resistance to abiotic stress. Therefore, not all modern cultivars are necessarily more resistant to foliar diseases than older genotypes. For instance, the two modern cultivars, 'Settler CL' (2009) and 'Overland' (2007), have been described as susceptible to foliar diseases [68].

The size and weight of the grains remained constant over time within each fungicide treatment, indicating that the historical improvements in yield might be due to other factors such as higher tillering capacity and more spikes or grains per unit area [7]. However, the production of smaller and lighter grains, together with poorer plant health, in the absence of fungicide, explains the considerable differences in yield across fungicide treatments.

The GPC of wheat is critical for the end-use quality of flour. In general, a higher concentration is desirable. Nevertheless, the GPC of whole wheat flour was found to decrease over the period studied. Several studies have reported similar relationships between landraces and older cultivars when compared to modern wheat releases, attributing this trend to the dilution effect of the increasing yield [13, 17, 59, 60]. However, in this study, the declining concentration of protein was only partially explained by the dilution effect of yield, as evidenced by the significant partial correlation between registration year and protein when controlling for yield. To account for the decreasing GPC, plant breeders have successfully selected for cultivars with enhanced protein functionality, thus improving the end-use quality of flour in modern wheat releases [7]. As was documented in this study, the effect of fungicide on GPC has been reported as dependent on environment with an inverse relationship to yield [68, 69, 70].

The accumulation of Cd in grains of bread wheat is a complex trait and has not been clearly associated with plant height [34]. On the one hand, because the competition of stems and leaves with grains as Cd sinks, an increasing trend over time of grain Cd would be expected as modern wheat cultivars tend to be shorter than their historical counterparts [71, 72]. On the other hand, since other agronomic parameters besides plant height, such as the number of stems and spikes, also negatively correlate with grain Cd, modern wheats with a higher tillering capacity would be expected to present lower grain Cd than older cultivars. Therefore, since currently only one reduced height gene (Rht8) has been associated with grain Cd, this study did not pursue the relationship between plant height on grain Cd. However, a study evaluating the relationship between historical dynamics of aboveground biomass partitioning variation and grain Cd in a panel of cultivars from the Great Plants would significantly contribute to the field.

Due to its frequent consumption, wheat is recognized as a major source of dietary Cd. Presently, an increasing trend of the grain Cd concentration in modern wheat cultivars is reported in the absence of fungicide, suggesting that the ability of wheat to accumulate Cd in grains might be increasing over time. Because of the considerable soil Cd concentration at the location studied (ENREC), most samples were above the maximum limit of Cd in China (0.1 mg kg$^{-1}$) and below that in Europe (0.2 mg kg$^{-1}$) [73, 74]. However, concentrations of up to 0.58 mg kg$^{-1}$ have been reported for wheat grown in the Great Plains [9].

The implications of an increasing grain Cd trend gain relevance when the constant contamination of cropland with Cd and the distribution of this heavy metal within the grain are considered [75, 76]. Whereas most micronutrients accumulate in the bran and embryo, Cd distributes throughout the grain [76]. Therefore, while during the production of white flour, most micronutrients are removed with the bran, white flour retains Cd [77], thereby exposing the general population consuming flour-derived food products to an increasingly richer and potentially dangerous source of this heavy metal.

The application of fungicide effectively reduced the grain Cd concentration of all samples and attenuated the increasing temporal trend. In locations with elevated concentration of soil

Cd, farmers must adopt methods to minimize the accumulation of Cd in grains. The use of soil amendments, such as organic matter, has proven effective in mitigating the accumulation of Cd in wheat by limiting the phytoavailability of Cd in soil [78]. Here, controlling disease pressure through the application of fungicide significantly decreased grain Cd. Thus, modern agronomic practices to reduce biotic stress of plants could help control the accumulation of Cd in grains. Nevertheless, understanding the molecular basis of the low grain Cd trait is still imperative.

As foliar fungal pathogens compete for nutrients with plants, the level of infection and types of fungicide used can affect the partitioning of carbohydrates, nitrogen, and micronutrients towards the grains [79–81]. Here, the application of fungicide (metconazole) efficiently reduced disease pressure, resulting in increased yield and GPC while simultaneously diluting the mineral elements in the grains when compared with untreated plants. However, this effect of fungicide should not be generalized as other active compounds could yield different results. For instance, fungicides based on strobilurins and carboxamides, besides controlling diseases, impart physiological effects such as delaying senescence [82]. Thus, these could allow for a greater deposition of carbohydrates or micronutrients in grains.

The genotype susceptibility to foliar diseases such as leaf and stripe rust could also affect the concentration of Cd and micronutrients in grains. However, since most of the cultivars included in this study were susceptible to foliar diseases, no evident relationship between genotype susceptibility mineral elements in grains was found.

Phytate, P, Mg, and Mn remained unchanged over time, suggesting that wheat strictly regulates the concentration of these elements in grains because of their physiological importance for seedling health. Phytate is essential for the physiology of wheat seeds, serving as the major reservoir of P and micronutrients to support seedling development [83]. During seed germination, P, Mg, and Mn are essential for the rapid production of ATP, carbohydrate partitioning, and photosynthesis required for the establishment and vigor of seedlings [84–87]. Nevertheless, under the biotic stress posed by disease pressure, the synthesis of phytate was impaired as evidenced by lower concentrations of phytate in non-treated plants.

The historical declining concentrations of Fe and Zn in wheat grains have been previously reported [9, 13, 14, 88]. In this study, the negative regression between registration year and Fe concentration became non-statistically significant when controlling for yield, indicating that this trend is primarily due to the dilution effect of the increasing yield [15]. However, the declining concentration of Zn under fungicide treatment remained statistically significant, suggesting that other factors besides yield dilution might drive this dynamic. Acknowledging this trend, efforts have been made towards the biofortification of wheat through plant breeding. For instance, the introgression of the high grain protein content allele (*Gpc-B1*), from emmer wheat (*Triticum turgidum* ssp. *dicoccoides)*, into bread wheat not only increases protein, but also increases Fe, and Zn concentrations of grains while having no detrimental effects on yield [23, 89–91]. Nevertheless, this allele is still rare among modern wheat cultivars [92]; thus, the development of cost-effective alternatives to counter the decling concentrations of Fe and Zn are urgent.

Although the decrease in mineral elements over time and across fungicide treatments has been attributed primarily to yield dilution, a PC analysis taking into account the differences in yield was still able to separate fungicide-treated samples from the non-treated samples. The separation was not due to differences in mineral elements, but was associated with grain size, weight, and GPC and probably other unidentified factors that distinguish fungicide treatments. For instance, the difference in grain size across fungicide treatments could partially explain the lower micronutrient concentration of treated plants, as the proportion of bran utilized during mineral elements analysis decreases for bigger grains [76]. Other agronomic

factors such as the number of grains per head and their position in the spikes, have also shown to affect the concentration of micronutrients [93].

In conclusion, although the grain yield increased steadily since 1933 without detrimental effects on seed phytate, Mg, and Mn concentrations, penalties on the concentration of essential nutrients such as protein, Fe, and Zn were evident. Perhaps most importantly, an unexpected increasing concentration of the heavy metal Cd, especially under biotic stress, was found, representing a possible threat to public health. Going forward, breeders will have to be vigilant for the selection of low Cd alleles while concurrently selecting for high yielding varieties. Although fungicide significantly decreased the concentration of all mineral elements that are important to human health and nutrition, the application of fungicide also effectively reduced grain Cd of all cultivars observed in this study. Therefore, fungicide application could serve as a useful agronomic practice for farmers to increase yield and economic returns while effectively limiting grain Cd concentrations.

## Supporting information

**S1 Fig. Scatter plot comparison of the leaf health score in the presence and absence of fungicide from 2017 (2018 data not available).**
(DOCX)

**S2 Fig.** Linear regressions of A) grain diameter and B) thousand kernels weight (TKW) by registration year for a historic set of 18 wheat cultivars released between 1933 and 2013, which were grown in a split-plot field experiment in the presence or absence of fungicide application at the Eastern Nebraska Research and Education Center in 2017 and 2018. The regression lines were adjusted for growing year effects and the shaded area around each regression line represents the 95% confidence interval. The landraces 'Turkey' (1874) and 'Kharkof' (1900) were not included in the regression analysis.
(DOCX)

**S1 Table. Pearson correlation coefficients between the economic traits, grain dimensions, phytate, micronutrients, and cadmium of wheat grains across growing years (2017–2018) in the presence and absence of fungicide.**
(DOCX)

**S2 Table. Raw data for response variables across two growing years and fungicide treatments.**
(XLSX)

## Acknowledgments

The authors are grateful to Dr. Robert Graybosch, U.S. Department of Agriculture-Agricultural Research Service (USDA-ARS) (retired), and Lori Divis, USDA-ARS, for their technical support.

## Author Contributions

**Conceptualization:** Hollman Motta-Romero, Devin J. Rose.

**Data curation:** Hollman Motta-Romero.

**Formal analysis:** Hollman Motta-Romero.

**Funding acquisition:** Devin J. Rose.

**Investigation:** Hollman Motta-Romero, Ferdinand Niyongira.

**Methodology:** Hollman Motta-Romero, Ferdinand Niyongira.

**Project administration:** Jeffrey D. Boehm, Jr., Devin J. Rose.

**Resources:** Jeffrey D. Boehm, Jr., Devin J. Rose.

**Supervision:** Devin J. Rose.

**Visualization:** Hollman Motta-Romero, Jeffrey D. Boehm, Jr.

**Writing – original draft:** Hollman Motta-Romero.

**Writing – review & editing:** Hollman Motta-Romero, Ferdinand Niyongira, Jeffrey D. Boehm, Jr., Devin J. Rose.

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
