## [Decision Letter · Decision Letter 0]

17 Aug 2020

PONE-D-20-20368

Effects of Foliar Fungicide on Yield, Micronutrients, and Cadmium in Grains from Historical and Modern Hard Winter Wheat Genotypes

PLOS ONE

Dear Dr. Rose

Thank you for submitting your manuscript to PLOS ONE. After careful consideration, we feel that it has merit but does not fully meet PLOS ONE’s publication criteria as it currently stands. Therefore, we invite you to submit a revised version of the manuscript that addresses the points raised during the review process.

I received comments from the advisers on revised version of your manuscript “Effects of Foliar Fungicide on Yield, Micronutrients, and Cadmium in Grains from Historical and Modern Hard Winter Wheat Genotypes“, which you submitted to PlosONE. Both reviewers agreed that the manuscript is well written, but some weaknesses in the experimental design i.e. only one fungicide application per growing season, selection of the cultivars and the evaluation of the Cd data raise serious worries. Nevertheless, after careful reading of the manuscript I decided to give you an opportunity to publish the current work, in case if you will be able to respond to reviewer questions and if you are prepared to incorporate major revisions. When preparing revised manuscript, you are asked to carefully consider the reviewer comments which can be found below, and submit a list of detailed and itemized responses to the comments.

We look forward to receiving your revised manuscript.

Kind regards,

Dragan Perovic, Ph.D

Academic Editor

PLOS ONE

Additional Editor Comments:

Dear Dr. Rose,

I received comments from the advisers on revised version of your manuscript “Effects of Foliar Fungicide on Yield, Micronutrients, and Cadmium in Grains from Historical and Modern Hard Winter Wheat Genotypes“, which you submitted to PlosONE. Both reviewers agreed that the manuscript is well written, but some weaknesses in the experimental design i.e. only one fungicide application per growing season, selection of the cultivars and the evaluation of the Cd data raise serious worries. Nevertheless, after careful reading of the manuscript I decided to give you an opportunity to publish the current work, in case if you will be able to respond to reviewer questions and if you are prepared to incorporate major revisions. When preparing revised manuscript, you are asked to carefully consider the reviewer comments which can be found below, and submit a list of detailed and itemized responses to the comments.

With kind regards

Dragan

Journal Requirements:

"This project was partially supported by the Nebraska Agricultural Experiment Station with funding from the Hatch Multistate Research capacity funding program (Accession Number 224073) from the USDA National Institute of Food and Agriculture. The funders had no role in study design, data collection and analysis, decision to publish, or preparation of the manuscript."

Reviewers' comments:

Reviewer's Responses to Questions

**Comments to the Author**

1. Is the manuscript technically sound, and do the data support the conclusions?

Reviewer #1: Yes

Reviewer #2: Partly

2. Has the statistical analysis been performed appropriately and rigorously? 

Reviewer #1: No

Reviewer #2: No

3. Have the authors made all data underlying the findings in their manuscript fully available?

Reviewer #1: Yes

Reviewer #2: Yes

4. Is the manuscript presented in an intelligible fashion and written in standard English?

Reviewer #1: Yes

Reviewer #2: Yes

5. Review Comments to the Author

Reviewer #1: The results are interesting, especially with respect to the trend of Cd concentration in winter wheat kernel over time and effect of Caramba fungicide on it. However, there are two points which should be clarified:

1. The criteria used to choose this particular set of 20 cultivars included in the experiment (listed in Table 1) are not clear, since mineral composition of wheat kernels adapted to that area (Great Plains USA) are well studied for over 200 hard winter wheat cultivars, current and historical, as shown in the article cited under no. 8 (Guttieri et al., 2015). Moreover, for example, genotypes under no. 13 and 14 are from 1983 and 1984 respectively (two consecutive years) and next is from 1998 even though between 1984 and 1998 eight other genotypes were released. Also, the year of release is not consistent with some previous publications, for example for Red Chief.

2. The results in the submitted paper do not take into account differences in the size of the grain (1000 grain weight) which can very up to about 20%. Therefore, the proportion of bran can be quite variable in different wheat cultivars. That can explain significant decrease in Fe, Zn and GPC content over registration years, for example. Besides, the majority of analyzed components are located in the bran which is usually removed during milling.

There are some additional suggestion:

1. Line 34 - since GPC appears for the first time in the text explain the acronym

2. Line 36 - since only 2 years are in question (and for some analyses only one), the term "period" sounds inappropriate

3. Line 168 - in 2 g of kernels how many of them did you have for different genotypes? How variable is their size?

4. Line 183 - missing space between 40 and units

5. Lines 272-275 - since wheat genotypes are not well distributed over registration years, the regression of response variables against them does not seem appropriate

6. Lines 366-376 - text needs revision.

7. Line 399 - P is not micronutrient but macronutrient

8. Line 524 - instead if "is" should stand "in"

Reviewer #2: The authors present the first comprehensive study on micronutrient and cadmium (Cd) concentrations in hard winter wheat (Triticum aestivum L.) in relation to the year of cultivars release in the presence and absence of fungicide application.

The manuscript is well-structured and written and mostly cites the relevant literature. However, I see some weaknesses in the experimental design (only one fungicide application per growing season, selection of the cultivars) and in the evaluation particularly of the Cd data, which led to alarmist conclusions. I recommend that the manuscript should not be published in PLOS One in the present form.

The introduction is focused on the scientific problem targeted. The general experimental design and the methods applied are mostly adequate. The study is based on only 20 cultivars, which I would consider the very low end suitable for a trend analysis. The cultivars have been selected by their historic importance for wheat production and contribution to the pedigrees of elite cultivars mainly grown in the Great Plains. However, four out of 20 cultivars/landraces released 1870, 1900, 1933, and 1940 represent half of the analyzed period. Hence, these four cultivars have a very strong influence on outcomes of the trend calculations. That holds also true for the selection of some of the outermost cultivars with respect to disease resistance (‘Kharkhof’, 1900, healthy; ‘Settler CL’ 2008, and ‘Freeman’ 2013, highly susceptible, cf. [459-463]). The author’s data show that yield increase over time with fungicides is stronger that without fungicides. This is not supported by the literature for trials where historic and modern cultivars are cultivated together and should therefore have been discussed in detail. Unfortunately that has not been done. Thus, the outcome could either reflect the breeding history in the Great Plains or just the trend of the selected cultivars, see above. Other authors (e.g. Ahlemeyer & Friedt 2011, Voss-Fels et al. 2019) found clearly improved resistance in wheat leading to higher yield increase without fungicides because of improved resistance which the authors confirmed in the introduction [55, without reference].

Secondly, only one fungicide application per growing season was carried out. That is in my regard very critical as the title of the study “The effects of foliar fungicide…” claims to analyze primarily the effects of foliar fungicides. One fungicide application is certainly not sufficient to control fungal pathogens effectively. It might resemble common agricultural practice in the Great Plains or extensive cultivation. Accordingly, an intense treatment, which would reflect the effects of foliar fungicides without confounding fungal pathogens is missing. The resulting problems are reflected in the data, e.g. cultivars ‘Kharkof’ and ‘Lancer’ respond negatively to fungicide application), the disease pressures between growing years differ strongly even with fungicides. Moreover, the authors recorded plant health, which could have at least shown the effectiveness of the fungicide treatments, only in growing season 2017. Why?

In general, the authors conducted a rigorous data analysis. I agree that linear regression is mostly used to analyze breeding progress of yield and other variables over time. However, one major factor of Cd concentration in grains is plant height (e.g. Kubo et al. 2008, Arduini et al. 2018, Payandeh et al. 2018) which did not change linear, particularly not in the panel of the selected cultivars. Although it is an easy and very often measured parameter it's not considered here. It is obvious from the provided data that the increase in Cd uptake occurred in cultivars released after the 60s along with the strong decrease in plant height and remained largely constant since. Accordingly, there would be increasing linear trend in Cd concentration in grains in the last 50 years and such, the major alarmist statement would not be supported.

References

Ahlemeyer, J., & Friedt, W. (2011). Progress in winter wheat yield in Germany-what's the share of the genetic gain?. Tagungsband der 61. Jahrestagung der Vereinigung der Pflanzenzüchter und Saatgutkaufleute Österreichs, 23-25 November 2010, Raumberg-Gumpenstein, Österreich. Ertrag vs. Qualität bei Getreide, Öl und Eiweisspflanzen. Wheat stress, 19-24.

Arduini I, Masoni A, Mariotti M, Pana S, Ercoli L. Cadmium uptake and translocation in durum wheat varieties differing in grain-Cd accumulation. Plant Soil Environ. 2018;60(1):43-9. doi: 10.17221/416/2013-pse.

Kubo, K., Watanabe, Y., Oyanagi, A., Kaneko, S., Chono, M., Matsunaka, H., ... & Fujita, M. (2008). Cadmium concentration in grains of Japanese wheat cultivars: genotypic difference and relationship with agronomic characteristics. Plant production science, 11(2), 243-249.

Payandeh, K., Jafarnejadi, A., Gholami, A., Shokohfar, A., & Panahpor, E. (2018). Evaluation of Cadmium Concentration in Wheat Crop Affected by Cropping System. Jundishapur Journal of Health Sciences, 10(2).

Voss-Fels, K. P., Stahl, A., Wittkop, B., Lichthardt, C., Nagler, S., Rose, T., ... & Ballvora, A. (2019). Breeding improves wheat productivity under contrasting agrochemical input levels. Nature plants, 5(7), 706-714.

6. PLOS authors have the option to publish the peer review history of their article (what does this mean?). If published, this will include your full peer review and any attached files.

Reviewer #1: No

Reviewer #2: No

---

## [Author Response · Author response to Decision Letter 0]

28 Sep 2020

Response to reviewers’ comments on PONE-D-20-20368

Reviewer 1

Q1. The criteria used to choose this particular set of 20 cultivars included in the experiment (listed in Table 1) are not clear, since mineral composition of wheat kernels adapted to that area (Great Plains USA) are well studied for over 200 hard winter wheat cultivars, current and historical, as shown in the article cited under no. 8 (Guttieri et al., 2015). Moreover, for example, genotypes under no. 13 and 14 are from 1983 and 1984 respectively (two consecutive years) and next is from 1998 even though between 1984 and 1998 eight other genotypes were released. Also, the year of release is not consistent with some previous publications, for example for Red Chief.

A1. Thanks for such an insightful comment, especially that regarding the inconsistency of the release years reported as some of these were indeed incorrect in the previous submission. Although we agree that having a larger panel of samples more homogeneously distributed along the time period studied would be the ideal, the landraces and elite cultivars included in this study effectively represent the genetic diversity and production history of the Great Plains. These cultivars significantly contributed to the pedigree of recent leading cultivars and occupied considerable acreage at their time. Moreover, this set of cultivars were selected due to their known adaptation to the location of the study. In studies looking at temporal trends and genetic gains, it is imperative to include samples adapted to the region of interest and known to perform at their best to avoid confounding factors. Since these samples have been employed by our research groups across multiple years, these are known to perform at their best under these climate conditions, despite the considerable soil cadmium concentration at the research center. Some of this language has been added to the text (lines 141-144).

To correctly report the release year for these samples, these data were cross-referenced with that reported in the Genetic Resources Information System for Wheat and Triticale (GRIS). From this comparison, the release years of the cultivars ‘Turkey’, ‘Anton’, and ‘Settler CL’ were corrected from 1870, 2007, and 2008 to 1874, 2008, and 2009, respectively. In the particular case of ‘Red Chief’, it is worth mentioning that multiple cultivars appear registered with the same or similar names. However, the one included in this study was indeed released in 1940. 

In the revision, because of the outlying nature of the release year of the landraces (1874, 1900) in the context of the period studied (1933-2013), these samples were treated as references of the improvement of elite wheat lines as a result of plant breeding. Therefore, these samples were not considered for the regression analyses in the revision. (lines 144-147, 251-255, and captions for figures and Table 2).

While the mineral composition of wheat kernels in the Great Plains has been reported previously (as noted by the reviewer), the effect of fungicide treatment has not. As we report in this manuscript, the effect of fungicide is significant for several mineral elements. This is an important contribution to knowledge and the literature.

Q2. The results in the submitted paper do not take into account differences in the size of the grain (1000 grain weight) which can very up to about 20%. Therefore, the proportion of bran can be quite variable in different wheat cultivars. That can explain significant decrease in Fe, Zn and GPC content over registration years, for example. Besides, the majority of analyzed components are located in the bran which is usually removed during milling.

A2. This is an excellent point by this reviewer. Indeed, most micronutrients concentrate in the bran, mainly the aleurone layer. Therefore, any variation in kernel size across registration years or fungicide treatments could help explain the effects of time or fungicide reported in this work. To address this concern, we measured grain diameter and thousand kernels weight (TKW) for all the samples included in this study. These variables were included in the ANOVA table (Table 2) and the new Figure S2. No significant trends were observed over time in the diameter or weight of kernels in the presence (2.7 mm; 28 g) or absence (2.6 mm; 26 g) of fungicide. Therefore, changes in the seed size (and therefore bran proportion) do not help explain the temporal dynamics reported in this study. Nevertheless, in the absence of fungicide, smaller and lighter grains were produced compared to with fungicide, which would partially explain the elevated element concentration in these samples compared with the fungicide-treated samples. (lines 171-173, 344-348, 489-493).

Q3. Additional suggestions.

Q3.1. Line 34 - since GPC appears for the first time in the text explain the acronym

A3.1. Corrected. 

Q3.2. Line 36 - since only 2 years are in question (and for some analyses only one), the term "period" sounds inappropriate.

A3.2. In this case, we refer to the 1933-2013 period.

Q3.3. Line 168 - in 2 g of kernels how many of them did you have for different genotypes? How variable is their size?

Q3.3. Please refer to the response for comment #2. Neither grain diameter nor thousand kernels weight significantly regressed with registration year. However, significantly smaller and lighter grains were produced in the absence of fungicide. 

Q3.4. Line 183 - missing space between 40 and units

A3.4. Corrected. 

Q3.5. Lines 272-275 - since wheat genotypes are not well distributed over registration years, the regression of response variables against them does not seem appropriate.

A3.5. For the revision, the landraces ‘Turkey’ (1874) and ‘Kharkof’ (1900) were included as references of the improvement of elite cultivars due to plant breeding. However, these landraces were not included in the regression analyses due to the outlying nature of their release years in the context of the period studied (1933-2013). Please also refer to the response to comment #1.

Q3.6. Lines 366-376 - text needs revision.

A3.6. Corrected. This comment refers to lines 376-377 in the revision. 

Q3.7. Line 399 - P is not micronutrient but macronutrient

A3.7. Corrected. 

Q3.8. Line 524 - instead if "is" should stand "in"

A3.8. Corrected.

Reviewer 2

The manuscript is well-structured and written and mostly cites the relevant literature. However, I see some weaknesses in the experimental design (only one fungicide application per growing season, selection of the cultivars) and in the evaluation particularly of the Cd data, which led to alarmist conclusions. I recommend that the manuscript should not be published in PLOS One in the present form.

Q1. The introduction is focused on the scientific problem targeted. The general experimental design and the methods applied are mostly adequate. The study is based on only 20 cultivars, which I would consider the very low end suitable for a trend analysis. The cultivars have been selected by their historic importance for wheat production and contribution to the pedigrees of elite cultivars mainly grown in the Great Plains. However, four out of 20 cultivars/landraces released 1870, 1900, 1933, and 1940 represent half of the analyzed period. Hence, these four cultivars have a very strong influence on outcomes of the trend calculations. That holds also true for the selection of some of the outermost cultivars with respect to disease resistance (‘Kharkhof’, 1900, healthy; ‘Settler CL’ 2008, and ‘Freeman’ 2013, highly susceptible, cf. [459-463]).

A1. We appreciate this comment and that of reviewer 1 regarding the distribution of samples across the years. As mentioned above, these cultivars were selected based on their known adaptability to the region, and effective representation of the genetic diversity and production history of the Great Plains. However, in this revision, because of the outlying nature of the release year of the landraces (1874, 1900) in the context of the period studied (1933-2013), these samples were treated as references of the improvement of elite wheat lines as a result of plant breeding and were not considered for the regression analyses. (lines 144-147, 251-255, and captions for figures and Table 2).

We agree that having a larger panel of samples would be the ideal. However, we do report significant trends over time for several nutrients, which indicates we had enough power to detect trends for at least some nutrients. Our results also report the effect of fungicide on the micronutrient concentration of wheat cultivars adapted to the Great Plains, which will be a new and important contribution to the literature. 

The comments about the disease resistance of the panel selected are addressed in the following response as the next comment is related to the effect of plant health on the results presented in the manuscript. 

Q2. The author’s data show that yield increase over time with fungicides is stronger that without fungicides. This is not supported by the literature for trials where historic and modern cultivars are cultivated together and should therefore have been discussed in detail. Unfortunately that has not been done. Thus, the outcome could either reflect the breeding history in the Great Plains or just the trend of the selected cultivars, see above. Other authors (e.g. Ahlemeyer & Friedt 2011, Voss-Fels et al. 2019) found clearly improved resistance in wheat leading to higher yield increase without fungicides because of improved resistance which the authors confirmed in the introduction [55, without reference].

A2. We agree and are aware of the multiple reports on the increasing disease resistance of wheat. We also understand that our results contrast with previous reports where higher estimates are reported in the absence than in the presence of fungicide (Martens et al., 2014; Voss-Fels, et al., 2019). In the cited cases, this has been attributed to the lower benefit from fungicide application of modern resistant cultivars, which in turn flattens the slope of a yield-vs-registration year regression under fungicide application when compared to its absence, where modern cultivars would clearly out yield historical varieties. Nevertheless, in the case of the great plains, particularly in the cultivars studied, not all modern cultivars were necessarily more resistant to foliar diseases than older genotypes. For instance, in our panel, the two modern cultivars, ‘Settler CL’ (2009) and ‘Overland’ (2007), have been described as susceptible to foliar diseases, thus benefiting from fungicide as much as older cultivars. (lines 479-488)

Q3. Only one fungicide application per growing season was carried out. That is in my regard very critical as the title of the study “The effects of foliar fungicide…” claims to analyze primarily the effects of foliar fungicides. One fungicide application is certainly not sufficient to control fungal pathogens effectively. It might resemble common agricultural practice in the Great Plains or extensive cultivation. Accordingly, an intense treatment, which would reflect the effects of foliar fungicides without confounding fungal pathogens is missing. The resulting problems are reflected in the data, e.g. cultivars ‘Kharkof’ and ‘Lancer’ respond negatively to fungicide application, the disease pressures between growing years differ strongly even with fungicides. Moreover, the authors recorded plant health, which could have at least shown the effectiveness of the fungicide treatments, only in growing season 2017. Why?

A3. The effect of fungicide on the selected economic and nutritional variables was chosen to be studied under a single application to mimic the common agronomical practice in the region. As the wheat growing season of the Great Plains is characterized by a short grain-filling duration, a single fungicide application is the most economically viable option (Wegulo et at., 2011). We agree that multiple applications of fungicide would indeed reduce confounding fungal pathogen factors in the study. However, it would not realistically represent field conditions for the region. (lines 162-165)

Unfortunately, plant health data were not recorded for 2018; hence, it is not reported in the manuscript. We consider that the effectiveness of a single fungicide application is effectively represented by the improvements in yield and plant health reported (2017). These same cultivars were planted in 2019 and 2020. The 2019 season was characterized by a higher disease pressure than 2020 due to high precipitations late in the season. The Plant health data from these years (see figures in Response to Reviewers document at the end of the PDF file) support that a single application effectively improves the leaf health score of most cultivars and that a greater benefit is achieved under higher disease pressure. However, these years are not included in the manuscript as mineral element composition data are not available. 

Figure: Scatter plot comparison of the leaf health score in the presence and absence of fungicide from the 2019 and 2020 seasons (see Response to Reviewers document within the PDF file). 

Q4. In general, the authors conducted a rigorous data analysis. I agree that linear regression is mostly used to analyze breeding progress of yield and other variables over time. However, one major factor of Cd concentration in grains is plant height (e.g. Kubo et al. 2008, Arduini et al. 2018, Payandeh et al. 2018) which did not change linear, particularly not in the panel of the selected cultivars. Although it is an easy and very often measured parameter it's not considered here. It is obvious from the provided data that the increase in Cd uptake occurred in cultivars released after the 60s along with the strong decrease in plant height and remained largely constant since. Accordingly, there would be increasing linear trend in Cd concentration in grains in the last 50 years and such, the major alarmist statement would not be supported.

A4. We agree that extensive research on the dynamics of Cd in durum wheat suggests that grain Cd concentration negatively correlates with plant height. As the leaves and stem of wheat compete as Cd sinks during grain filling, less Cd reaches the grains in taller cultivars (Harri & Taylor, 2013; Perrier et al., 2016). Nevertheless, the accumulation of Cd in grains of wheat is a complex trait and has not been molecularly associated with plant height yet. 

The gene Cdu1 controlling the root-to-shoot Cd translocation in durum wheat has allowed breeders to introduce the low-Cd trait in modern durum wheat cultivars (Knox et al., 2009). Nevertheless, the knowledge on durum wheat does not seem to directly translate to bread wheat. A homoeologous locus to Cdu1 in bread wheat has not been identified. Moreover, only one infrequent reduced height gene (Rht8) has been related to high grain Cd in bread wheat (Guedira et al., 2010; Knopf et al., 2008; Liu et al., 2019). For these reasons, and because of the intrinsically related nature of the release year and plant height factors, that is, most historical wheats are full-stature lines and most modern cultivars are semi-dwarfs, no comparisons based on their plant height were pursued in this work. 

We agree that the relationship between plant height and grain Cd is worth exploring and that future studies on the relationship between historical dynamics of aboveground biomass partitioning variation and grain Cd in a large panel of cultivars would significantly contribute to the field. However, the goal of this study was to study the dynamic of cadmium concentration over time and the effect of fungicide application in bread wheat. We consider that the results obtained are worth publishing as cadmium proves to behave differently to mineral elements such as iron and zinc, which are diluted by the increasing yield. 

In response to the “alarmist conclusions” statement by this reviewer, the conclusions on the temporal trend of cadmium are softened and less distressing (lines 523-538). Although an increasing concentration of grain cadmium is still reported, the application of fungicide effectively reduced the range of concentration and attenuated the increasing temporal trend. Therefore, our results suggest that reducing the biotic stress through the application of fungicide could help to control grain Cd concentration. However, we acknowledge that understanding the molecular basis of the low grain cadmium trait in bread wheat is still imperative. (please see modified text lines 38-40, 101-124, 358-366, 506-534, 571-579).

References

Ahlemeyer, J., & Friedt, W. (2011). Progress in winter wheat yield in Germany-what's the share of the genetic gain?. Tagungsband der 61. Jahrestagung der Vereinigung der Pflanzenzüchter und Saatgutkaufleute Österreichs, 23-25 November 2010, Raumberg-Gumpenstein, Österreich. Ertrag vs. Qualität bei Getreide, Öl und Eiweisspflanzen. Wheat stress, 19-24.

Arduini I, Masoni A, Mariotti M, Pana S, Ercoli L. Cadmium uptake and translocation in durum wheat varieties differing in grain-Cd accumulation. Plant Soil Environ. 2018;60(1):43-9. doi: 10.17221/416/2013-pse.

Guedira M, Brown-Guedira G, Sanford D, Sneller C, Souza E, Marshall D. Distribution of Rht genes in modern and historic winter wheat cultivars from Eastern and Central USA. Crop Sci. 2010; 50: 1811–1822

Harris N, Taylor G. Cadmium uptake and partitioning in durum wheat during grain filling. BMC Plant Biol. 2013; 13: 103.

Knopf C, Becker H, Ebmeyer E, Korzun V. Occurrence of three dwarfing Rht genes in German winter wheat varieties. Cereal Res. Commun. 2008; 36: 553-560.

Knox, R., Pozniak, C., Clarke, F., Clarke, J., Houshmand, S., & Singh, A. (2009). Chromosomal location of the cadmium uptake gene (Cdu1) in durum wheat. Genome, 52(9), 741-747.

Kubo, K., Watanabe, Y., Oyanagi, A., Kaneko, S., Chono, M., Matsunaka, H., ... & Fujita, M. (2008). Cadmium concentration in grains of Japanese wheat cultivars: genotypic difference and relationship with agronomic characteristics. Plant production science, 11(2), 243-249.

Liu C, Guttieri M, Waters B, Eskridge K, Baenziger P. Selection of bread wheat for low grain cadmium concentration at the seedling stage using hydroponics versus molecular markers. Crop Sci. 2019; 59: 945-956.

Martens, G., Lamari, L., Grieger, A., Gulden, R., & McCallum, B. (2014). Comparative yield, disease resistance and response to fungicide for forty-five historic Canadian wheat cultivars. Canadian Journal of Plant Science, 94(2), 371-381

Payandeh, K., Jafarnejadi, A., Gholami, A., Shokohfar, A., & Panahpor, E. (2018). Evaluation of Cadmium Concentration in Wheat Crop Affected by Cropping System. Jundishapur Journal of Health Sciences, 10(2).

Perrier, F., Yan, B., Candaudap, F., Pokrovsky, O., Gourdain, E., & Meleard, B. (2016). Variability in grain cadmium concentration among durum wheat cultivars: Impact of aboveground biomass partitioning. Plant and Soil, 404(1-2), 307-320.

Voss-Fels, K. P., Stahl, A., Wittkop, B., Lichthardt, C., Nagler, S., Rose, T., ... & Ballvora, A. (2019). Breeding improves wheat productivity under contrasting agrochemical input levels. Nature plants, 5(7), 706-714.

Wegulo, S., Zwingman, M., Breathnach, J., & Baenziger, P. (2011). Economic returns from fungicide application to control foliar fungal diseases in winter wheat. Crop Protection, 30(6), 685-692.

---

## [Decision Letter · Decision Letter 1]

26 Jan 2021

PONE-D-20-20368R1

Effects of Foliar Fungicide on Yield, Micronutrients, and Cadmium in Grains from Historical and Modern Hard Winter Wheat Genotypes

PLOS ONE

Dear Dr. Rose,

Thank you for submitting your manuscript to PLOS ONE. After careful consideration, we feel that it has merit but does not fully meet PLOS ONE’s publication criteria as it currently stands. Therefore, we invite you to submit a revised version of the manuscript that addresses the points raised during the review process.

Dear Dr. Rose,

Since the first revised version resulted in two opposite opinions, I was forced to invite the third one. Accordingly, the reviewing process was extended, and got a positive response.  I agreed with this opinion, therefore decided to give you an opportunity to publish the current work, in case if you will be able to respond to the third reviewer questions and if you are prepared to incorporate minor revisions. When preparing a revised manuscript, you are asked to carefully consider the third reviewer comments and submit a list of detailed and itemized responses to the comments.

With kind regards

Dragan

We look forward to receiving your revised manuscript.

Kind regards,

Dragan Perovic, Ph.D

Academic Editor

PLOS ONE

Additional Editor Comments (if provided):

Dear Dr. Rose,

Since the first revised version resulted in two opposite opinions, I was forced to invite the third one. Accordingly, the reviewing process was extended, and got a positive response.

I agreed with this opinion, therefore decided to give you an opportunity to publish the current work, in case if you will be able to respond to the third reviewer questions and if you

are prepared to incorporate minor revisions. When preparing a revised manuscript, you are asked to carefully consider the third reviewer comments and submit a list of detailed and

itemized responses to the comments.

With kind regards

Dragan

Reviewers' comments:

Reviewer's Responses to Questions

**Comments to the Author**

1. If the authors have adequately addressed your comments raised in a previous round of review and you feel that this manuscript is now acceptable for publication, you may indicate that here to bypass the “Comments to the Author” section, enter your conflict of interest statement in the “Confidential to Editor” section, and submit your "Accept" recommendation.

Reviewer #1: All comments have been addressed

Reviewer #3: All comments have been addressed

2. Is the manuscript technically sound, and do the data support the conclusions?

Reviewer #1: Yes

Reviewer #3: Yes

3. Has the statistical analysis been performed appropriately and rigorously? 

Reviewer #1: Yes

Reviewer #3: Yes

4. Have the authors made all data underlying the findings in their manuscript fully available?

Reviewer #1: Yes

Reviewer #3: Yes

5. Is the manuscript presented in an intelligible fashion and written in standard English?

Reviewer #1: Yes

Reviewer #3: Yes

6. Review Comments to the Author

Reviewer #1: (No Response)

Reviewer #3: Dear Devin J. Rose,

thank you for this very interesting study about macronutrients and Cadmium contamination of seeds. Cd can be added to the soil through natural and anthropogenic activities, fertilizers derived from phosphate rock. Authors conclude a relationship between biotic stress and the increased Cd content in the absence of fungicide treatment. Suggestions and remarks of two reviewers have been considered already, however, I suggest some minor changes.

Biotic stresses could arise by many reasons. If data to the resistance level against fungal diseases in the non fungicide treated control are available, please provide and discuss the relationship of infection to the Cd content. In lines 288 to 293 you mentioned, that "a rainy June allowed for higher disease pressure after flowering with leaf rust (Puccina triticina), bacterial leaf streak (Xanthomonas translucens), and fusarium head blight (Fusarium graminearum) as the main diseases observed". Please include some information about the resistance level of the cultivars, especially regarding yellow rust and leaf rust. Fungal (leaf) pathogens are strong sinks for nutrients, and probably for heavy metals.

Furthermore, fungicide treatment could not be generalized. In case of Metconazole as active compound of Caramba, a typical single-site fungicide has been used.

It should be discussed, that other compounds, e.g. strobilurins, could result in contradictive results due to a "greening effect".

7. PLOS authors have the option to publish the peer review history of their article (what does this mean?). If published, this will include your full peer review and any attached files.

Reviewer #1: No

Reviewer #3: **Yes: **Albrecht Serfling

---

## [Author Response · Author response to Decision Letter 1]

8 Feb 2021

Reviewer 3

Q: Thank you for this very interesting study about macronutrients and Cadmium contamination of seeds. Cd can be added to the soil through natural and anthropogenic activities, fertilizers derived from phosphate rock. Authors conclude a relationship between biotic stress and the increased Cd content in the absence of fungicide treatment. Suggestions and remarks of two reviewers have been considered already, however, I suggest some minor changes.

Biotic stresses could arise by many reasons. If data to the resistance level against fungal diseases in the non fungicide treated control are available, please provide and discuss the relationship of infection to the Cd content. In lines 288 to 293 you mentioned, that "a rainy June allowed for higher disease pressure after flowering with leaf rust (Puccina triticina), bacterial leaf streak (Xanthomonas translucens), and fusarium head blight (Fusarium graminearum) as the main diseases observed". Please include some information about the resistance level of the cultivars, especially regarding yellow rust and leaf rust. Fungal (leaf) pathogens are strong sinks for nutrients, and probably for heavy metals.

A: Thank you for these comments. As foliar fungal pathogens compete for nutrients with plants, the level of infection can affect the partitioning of carbohydrates, nitrogen, and micronutrients in the grains. Therefore, we agree that addressing the relationship between genotype susceptibility to foliar fungal diseases and grain micronutrients and Cd will strengthen this manuscript. In this revision, we report the genotype susceptibility to leaf and stripe rust in Table 1 as susceptible (S), moderately susceptible (MS), and moderately resistant (MR). To investigate the relationship between genotype susceptibility to foliar diseases and mineral elements in grains, the cultivars were divided into two groups: 1) S and 2) MS/MR. Then, the difference in grain mineral elements between fungicide treatments was compared between these two groups. No significant differences were found for Cd or any micronutrient (P > 0.05). However, since most of the cultivars included in this study were susceptible to foliar diseases, the size of the MS/MR group was very small (N=6 for leaf rust and N=3 for stripe rust). Therefore, although no evident relationship between genotype susceptibility to foliar diseases and mineral elements in grains was found in this study, additional research with a balanced experimental design would be more conclusive. This is now included in the Results and Discussion sections [lines 418-422, 579-582 (track changes); lines 412-416, 556-559 (clean)]. 

Q: Furthermore, fungicide treatment could not be generalized. In case of Metconazole as active compound of Caramba, a typical single-site fungicide has been used. It should be discussed, that other compounds, e.g. strobilurins, could result in contradictive results due to a "greening effect".

A: In this study, the application of fungicide (metconazole) efficiently reduced disease pressure, resulting in increased yield and GPC while simultaneously diluting the mineral elements in the grains when compared with untreated plants. However, this effect of fungicide should not be generalized as other active compounds could yield different results. For instance, fungicides based on strobilurins and carboxamides, besides controlling diseases, impart physiological effects such as delaying senescence. Thus, other fungicides could allow for a greater deposition of carbohydrates or micronutrients in grains and yield different results from those presented here. This discussion point has been added to the Discussion [lines 566-575 (track changes); lines 546-555 (clean)].

---

## [Editor Report · Decision Letter 2]

16 Feb 2021

Effects of Foliar Fungicide on Yield, Micronutrients, and Cadmium in Grains from Historical and Modern Hard Winter Wheat Genotypes

PONE-D-20-20368R2

Dear Dr. Rose,

We’re pleased to inform you that your manuscript has been judged scientifically suitable for publication and will be formally accepted for publication once it meets all outstanding technical requirements.

Kind regards,

Dragan Perovic, Ph.D

Academic Editor

PLOS ONE

Additional Editor Comments (optional):

Dear Dr. Rose,

it is my pleasure to accept your manuscript to be published in Plos ONE.

Thanks for your patience.

With kind regards

Dragan
---

## [Editor Report · Acceptance letter]

18 Feb 2021

PONE-D-20-20368R2 

Effects of Foliar Fungicide on Yield, Micronutrients, and Cadmium in Grains from Historical and Modern Hard Winter Wheat Genotypes 

Dear Dr. Rose:

I'm pleased to inform you that your manuscript has been deemed suitable for publication in PLOS ONE. Congratulations! Your manuscript is now with our production department. 

Kind regards, 

on behalf of

Dr. Dragan Perovic 

Academic Editor

PLOS ONE